# Advanced PSMA-PET/CT Imaging Parameters in Newly Diagnosed Prostate Cancer Patients for Predicting Metastatic Disease

**DOI:** 10.3390/cancers15041020

**Published:** 2023-02-06

**Authors:** Yaniv Yechiel, Yaly Orr, Konstantin Gurevich, Ronit Gill, Zohar Keidar

**Affiliations:** 1Department of Nuclear Medicine, Rambam Health Care Campus, Haifa 3109601, Israel; 2Rappaport Faculty of Medicine, Technion—Israel Institute of Technology, Haifa 3109601, Israel

**Keywords:** PSMA-PET/CT, prostate cancer, prostate PSMA tumor volume, metastases

## Abstract

**Simple Summary:**

Prostate cancer is among the most common malignancies in men worldwide. Many patients undergo a PSMA-PET/CT study for staging assessment. The aim of this retrospective study was to evaluate the relationship between advanced imaging parameters such as prostate PSMA tumor volume and the presence of metastatic disease in newly diagnosed prostate cancer patients undergoing PSMA-PET/CT for staging purposes. PSMA-PET/CT of 85 patients was analyzed, and these advanced imaging parameters were found to be statistically capable of assessing the likelihood of the presence of metastatic disease.

**Abstract:**

(1) Purpose: Recent studies indicate that advanced imaging parameters such as prostate PSMA tumor volume may have a value in predicting response to treatment of castration-resistant prostate cancer patients. In this study, we examine whether a relationship can be found between advanced imaging parameters such as prostate PSMA-TV and the presence of metastatic disease in newly diagnosed prostate cancer patients undergoing PSMA-PET/CT for staging purposes; (2) Methods: We retrospectively analyzed PET/CT studies of 91 patients with newly diagnosed prostate cancer. Prostate PSMA-TV was measured using the MIRADA-XD software. PET/CT results were recorded, as well as additional clinical parameters such as the Gleason score, etc.; (3) Results: Prostate PSMA-TV measurements were found to be able to significantly differentiate metastatic from the non-metastatic patient groups (13.7 vs. 5.5, *p*-value < 0.05). Overall, 54% percent of patients with levels of over 8.1 PSMA-TV had metastatic lesions found on their PSMA-PET/CT. A model based on this cutoff attained a sensitivity of 80%, a specificity of 68.3%, and a negative predictive value of 93.5% for identifying metastatic disease. Another bin model was found statistically capable of assessing the likelihood of the presence of metastatic disease with a *p*-value of 0.001; (4) Conclusions: Prostate PSMA-TV measurement has the potential to predict the presence of metastatic disease at staging and thus may impact further treatment decision and patient management.

## 1. Introduction

Prostate cancer is among the most common cancer in men worldwide, with an estimated 1,400,000 cases and 381,000 deaths annually [1]. The average age of a newly diagnosed prostate cancer is approximately 66 years of age [2]. Hence, prostate cancer is a major global healthcare challenge, compounded by an aging population and increasing frequency of diagnosis [3]. 

In the setting of the initial diagnosis, patients with high-risk localized prostate cancer undergo additional tests for staging purposes. For many years, conventional imaging modalities such as CT and bone scans were used. Bone scintigraphy was used mainly because of the high likelihood of skeletal involvement in prostate cancer. Current international guidelines recommend the use of CT, bone scintigraphy, or MR imaging for high-risk patients only. It should be noted that there are many approaches to risk assessment. However, these assays have insufficient sensitivity when staging men with high-risk localized prostate cancer [4], and clinicians rely on additional tools for severity stratification [5,6].

Namely, pathology results that are classified using the Gleason score were found to be the most significant when assessing the likelihood of advanced malignant disease [5]. However, the presence of metastatic disease remains the most important factor regarding patient outcomes. While the 5-year survival rate for prostate cancer patients without metastatic spread is nearly 100%, those with distant metastatic disease have 5-year survival rates of approximately 30% [3,7].

Hence, those patients with the high-risk localized disease need a better diagnostic method for identifying distant metastasis. In recent years, PSMA-PET/CT has been the preferred staging modality for both primary and recurrent prostate cancer due to its superior sensitivity and specificity [4,5,6,7,8,9,10,11,12,13,14,15], especially in assessing extra-skeletal involvement such as nodal metastasis. The superior accuracy of PSMA-PET/CT over conventional imaging for staging in high-risk patients may allow the identification of patients with otherwise occult distant metastatic disease. It could facilitate individualized multimodal treatment concepts, especially in the setting of oligometastatic disease [4]. Furthermore, the NCCN guidelines were updated to include this modality as the imaging modality of choice [16]. PSMA PET/CT has an evolving role in PSMA-targeting treatments in advanced disease (e.g., ^177^Lu-PSMA radioligand therapy) to evaluate target expression and, therefore, potentially predict the response before treatment initiation. High uptake at PSMA PET/CT is a prerequisite in selecting patients who may benefit from PSMA-directed radionuclide therapy [4]. However, it is important to mention some limitations regarding using this technology, such as high costs and availability (geographical and temporal meaning), as well as this radiotracer’s pitfalls, such as physiologic and other pathologic processes that can express PSMA, that imaging specialists should be familiar with.

PSMA is a transmembrane glycoprotein that is overexpressed in prostate cancer cells. Radiolabeled small molecules that bind with high affinity to their active extracellular center are the base of this imaging technique’s mechanism [7]. When performing PET/CT scans for evaluating prostate cancer patients, usually ^68^Ga-PSMA or ^18^F-PSMA are used. Notably, the ^18^F-PSMA tracer performs at least comparably to ^68^Ga-PSMA. Still, its longer half-life combined with its superior energy characteristics and non-urinary excretion overcomes some practical limitations of ^68^Ga-labelled PSMA targeted tracers—enabling excellent assessment of the prostate and its vicinity [16,17]. Additionally, ^18^F-PSMA is also thought to have higher sensitivity for low-grade lesions [18].

Recent studies indicate that advanced imaging parameters such as PSMA-derived tumor volume (PSMA-TV), total lesion PSMA (TL-PSMA), and PSMA total lesion quotient (PSMA-TLQ) measured using PSMA-PET/CT have a promising potential role in the assessment and treatment response prediction in castration-resistant prostate cancer patients [18,19]. These parameters are extracted from the PSMA-PET/CT data. Segmentation of each lesion is done semi-automatically by calculating each individual lesion over a specific threshold of the maximum local SUV. 

This study examines PET/CT data of newly diagnosed prostate cancer patients undergoing the test as part of their initial staging assessment. We investigated whether a relationship can be found between advanced imaging parameters and the presence of metastatic lesions found in PET/CT and evaluated if a measurable threshold differentiating metastatic vs. non-metastatic patients can be found.

## 2. Materials and Methods

Our study was a single-center retrospective cohort study in Rambam Health Care Campus in Israel, examining newly diagnosed prostate cancer patients who underwent PET/CT study with ^18^F-PSMA for staging assessment between January 2019 and December 2020. Overall, 91 patients met the inclusion criteria, six of which were excluded from the study due to the absence of prostatic tracer uptake (3) after a previous prostatectomy (2) or technically inappropriate PET/CT data (1).

^18^F-PSMA PET/CT acquisition and analysis: PET and contrast-enhanced CT (when not contraindicated) were acquired consecutively from head to the mid-thigh using a PET/CT system (Discovery 690, GE Healthcare, Milwaukee, WI, USA) approximately 60 min after the injection on average of 296 MBq (8 mCi) ^18^F-PSMA. The following parameters were used for CT imaging: pitch 1.375:1, gantry rotation time 0.7 s, 120 kVp, automatically adjusted current in the range of 100–650 mA and a 2.5 mm slice thickness. A contrast-enhanced CT scan was obtained 60 s after injection of 2 mL/kg of non-ionic contrast (Omnipaque 300; GE Healthcare). A PET scan followed in 3D acquisition mode for the same axial coverage. CT images were used for fusion with the PET data. PET images were reconstructed with CT attenuation correction using a 3D ordered subset expectation maximization (3D-OSEM).

All PET/CT studies were analyzed by a nuclear medicine specialist. All studies were reviewed retrospectively with knowledge of the patient’s clinical history and results of previous imaging studies. For each patient, the original PET/CT DICOM data was reviewed using the MIRADA-XD software (Mirada Medical Ltd., Oxford, UK). Using this platform, we semi-automatically calculated the following computerized parameters for each patient:

PSMA-TV (PSMA tumor volume)—calculated as the sum of all lesion volumes in the whole scanned body over a defined threshold of 42% of the maximum SUV in the lesion.

Prostate PSMA-TV—calculated as the sum of the lesion volume only within the prostate gland over the defined threshold of 42% of the maximum SUV in the lesion.

TL-PSMA (total lesion PSMA)—PSMA-TV × SUVmean measured in the prostate lesion.

PSMA-TLQ (total lesion quotient)—PSMA-TV/SUVmean measured in the prostate lesion.

Additional imaging information was gathered from the PET/CT, including the SUVmax and SUVmean values measured in the prostate gland. Other medical and demographic information, including age, initial PSA levels at diagnosis, Gleason score, patient clinical staging, treatments, and management, as well as clinical outcome, was recorded in a designated electronic case report form (eCRF).

All statistical tests applied in this retrospective analysis were performed using SPSS V.26. A standard T-test was used to evaluate whether traditional prognostic parameters, such as Gleason score and initial PSA and advanced imaging parameters, such as prostate PSMA-TV, differed significantly between the metastatic and non-metastatic cohorts. A *p*-value of <0.05 was defined as indicating a significant difference. Sensitivity, specificity, and positive and negative predictive values of the proposed prostate PSMA-TV threshold for differentiating metastatic from non-metastatic patients were calculated using med-calc software and were expressed as percentages. We used logistic regression to quantify the relationship between the main predictor variable (prostate PSMA-TV) and the response variable (presence of metastases).

The hospital ethics committee approved this retrospective study, and the patient’s informed consent was waived.

## 3. Results

The study population was comprised of 85 male patients with prostate cancer (Table 1). The mean age of the cohort was 72 years, with an average Gleason score of 7.39 for the entire cohort. The average follow-up time after the initial PET/CT was approximately 14 months; after initial disease assessment, during the follow-up period in which, 45 patients received either hormonal or radiation therapy, 16 patients underwent radical prostatectomy, and 17 patients were managed in a conservative active surveillance approach. Overall, five patients died during the follow-up period, and all of them had metastases discovered in the initial evaluation.

The two major sub-cohorts in our study were the metastatic and non-metastatic cohorts as defined by the PSMA PET/CT. Overall, 19 out of the 85 included patients were found to have metastatic lesions. The most frequent locations for metastases in our cohort were bones and lymph nodes, with ten patients having metastatic lesions in both locations, five patients had isolated lymph node metastases, and four had bone metastases only.

The only parameter that was found to be statistically different between the metastatic and non-metastatic groups was the Gleason score. It should be noted that the Gleason score of four patients from the metastatic group and five from the non-metastatic group could not be retrieved and thus was omitted from this analysis. Apart from the Gleason score, levels of neither SUVmax nor SUVmean were found as capable of significantly distinguishing between both groups.

In comparison, from the advanced imaging parameters measured, both prostate PSMA-TV and PSMA-TV measurements were found to be significantly different between both groups. PSMA-TLQ measurement differences gave a borderline significance with a *p*-value of exactly 0.05. Measurements and the level of significance between the two groups are detailed in Table 2:

Since PSMA-TV levels are inclusive of the metastatic lesion measurements, the contribution of this parameter to the current study design is minimal and has no further implications. Further analysis of prostate PSMA-TV distribution across the entire cohort (Figure 1) showed that 74% percent of the entire study population had measured prostate PSMA-TV levels under 6.5. At the same time, a relatively small number of outliers stretched the cohort mean value to 7.47.

The optimal binary classification point to differentiate between patients with or without metastases was examined, and the prostate PSMA-TV threshold of 8.1 was found as the most suitable cutoff to differentiate between these groups (Figure 2—ROC curve). We tested a classification system based on this cutoff aimed to predict the existence of metastatic disease based on prostate PSMA-TV alone. The results of this classification system are presented in Table 3 and Table 4.

Overall, this model attained a sensitivity of 80%, a specificity of 68.3%, and a negative predictive value of 93.5% for identifying patients with metastatic lesions. Six patients with metastases were not identified by this model. Importantly, four patients out of the five who died during the follow-up period of this study had levels over 8.1.

Lastly, we examined if a different model of stratifying the likelihood of metastatic disease can be established. Using logistic regression, we aimed to quantify the relationship between prostate PSMA-TV levels and the likelihood of the existence of metastatic lesions. Our model (presented in Table 5) identified three separate bins, classified by the likelihood of having metastatic lesions found in the PET/CT scan. For each bin, the likelihood of the presence of metastases increases 2.7-fold. The differences between each bin in this classification were found to be statistically significant using a Pearson Chi-Square test, with a *p*-value of 0.001.

## 4. Discussion

The advanced imaging parameters assessed in this study were previously examined only in correlation to patient outcomes of a single treatment course of prostate cancer [20]. To the best of our knowledge, this is the first study to assess these parameters and their correlation to metastatic disease in newly diagnosed patients. Since prostate cancer is such a prevalent disease in the aging male population, additional tools for identifying high-risk patients can be useful. We examined different approaches trying to provide physicians with an easy and useful tool for predicting metastatic disease.

Predicting the existence of metastatic lesions using traditional parameters is challenging. In our study, we found that even though the mean age, initial PSA level, Gleason score, and SUVmax were higher in the metastatic patient group, only the Gleason score was found to be significantly different between both study groups. This finding is in concordance with previous reports, showing that a high Gleason score does generally correlate with the existence of metastatic disease [21].

While Gleason scores of 8, 9, and 10 are all considered high-grade lesions, the average Gleason score of 8.2 in our metastatic group is slightly lower than expected for the metastatic cohort. Additionally, it is important to note that 15 out of 25 (60%) patients with a Gleason score >7 were found to be without metastatic disease at presentation, and 5 patients with Gleason ≤7 did have metastases on initial assessment. Thus, despite the fact that it was found to be statistically different between groups, in the real world, the Gleason score alone could probably not have provided the physicians with the necessary tool to accurately predict which patients will have metastatic disease. This could be explained by the small number of metastatic patients (*n* = 19) allowing for statistical variance or by the fact that the Gleason score of 4 metastatic patients was missing from their medical records.

While examining the new advanced imaging parameters, a couple of statistically significant differences were found. PSMA-TV was significantly different between the two groups. Still, this pattern was expected because the calculation of this parameter includes the measurement of all metastatic lesions in the body, as described in previous sections. On the contrary, prostate PSMA-TV was measured only in the prostate gland. Hence, the fact that it was found to be significantly elevated in the metastatic group indicates that the prostatic PSMA tumor volume increases in patients with metastatic disease independently from the measurement of the metastatic disease. Therefore, a reasonable assumption can be made that patients with elevated prostate PSMA-TV levels have a higher likelihood of developing metastatic disease.

Based on these results, we examined several different approaches in order to create a useful tool that may help physicians identify which patients are at high risk of developing metastatic lesions even though they do not have metastases in their initial PET/CT. This paper presents two models—one binary approach and another multi-bin approach.

For the binary model, we used a cutoff of 8.1 to differentiate the metastatic group from the non-metastatic group. In this model, all patients with measured prostate PSMA-TV values over 8.1 were classified as being at high risk of having metastatic lesions found in their PET/CT. Out of those who had levels over 8.1, more than 50% (13 out of 24) had metastases found in their initial PET/CT. Due to the correlation between the existence of metastases and high prostate PSMA-TV, even patients that did not have metastases found in their initial imaging may be considered at high risk for developing a metastatic disease in the future. This could mean a lot for physicians and patients in terms of more active and rigorous follow-up and treatment plans.

In contrast, those with prostate PSMA-TV levels lower than 8.1 can be considered as having a smaller chance of developing metastatic lesions. However, since 10% of patients in this group still had a metastatic disease found in their initial imaging, it is by no means a ‘magic bullet.’ In a larger cohort, this model might miss a considerable number of patients with metastases. Therefore, we believe that levels under 8.1 should not be used to rule out the likelihood of metastatic disease.

The main strength of this model is its ability to identify those patients with the highest risk of having metastatic disease. Accordingly, we suggest that prostate PSMA-TV levels over 8.1 are mainly useful as a ‘rule-in’ tool for stricter follow-up and treatment approaches in patients who do not have metastases found in their initial PET-CT but should be a part of a high-risk group for developing metastatic disease.

In contrast, the multi-bin system offers an approach that is more relevant for the entire population. In this model, the study population was divided into three separate bins according to the likelihood of the existence of metastatic lesions. When moving from each bin to the next, patients were 2.7 times more likely to have metastases than in the previous bin.

In the ‘low’ bin, 93% of patients did not have metastatic lesions found, and therefore patients with normal PET/CT results can be considered as ‘low risk’ and managed in a conservative manner. In the ‘medium’ bin, the likelihood of metastatic disease increases, but most patients do not have malignant disease. Our recommendation for this sub-group is to maintain routine treatment as would have been applied without the prostate PSMA-TV results. Lastly, over 73% (14 out of 19) of the metastatic patient in our cohort were found in the ‘high’ bin. In addition, almost 50% of the total patient in this group (14 out of 31) had metastases found, signaling that patients in this group should be considered as ‘high risk’ for metastatic disease and managed more aggressively even if their PET/CT results show localized disease only.

The main advantage of this model is its applicability to the entire cohort, giving the physician a tool to assess the likelihood of malignant disease for all patients undergoing PET/CT for staging purposes.

The major strength of our study is its novel approach to trying and using new advanced imaging parameters to assess disease severity and extent in prostate cancer patients. Since PET/CT is part of the routine workup for high-risk prostate cancer patients, this approach does not require additional resources from medical institutions and can aid physicians in patient management. Importantly, all measurements in this study were done semi-automatically, making the measurements less susceptible to bias. Lastly, all PSMA-PET/CT scans in this study used [^18^F] tracers, thus utilizing its excellent assessment of the prostate and superb sensitivity for localized lesions.

Our study has a few limitations. First, the analysis was performed retrospectively and is therefore prone to selection biases. Additionally, our results are based on a single center with a medium size cohort and thus cannot be generalized reliably into a larger population. Current results need to be confirmed in a large prospective study in order to estimate better the role of advanced imaging parameters such as prostate PSMA-TV in predicting metastatic disease in newly diagnosed prostate cancer patients. Moreover, the results correlating prostate PSMA-TV to the existence of metastases need to be verified with future research that will re-examine patients that were originally non-metastatic. To this purpose and due to the ongoing nature of this disease, a longer follow-up period of time is required. Lastly, comparison to other common staging methods, such as bone scintigraphy, CT scans, and MR imaging, may have added value to the significance of this study.

## 5. Conclusions

Prostate PSMA-TV may be a superior advanced imaging parameter compared to some traditional parameters in predicting the presence of metastatic disease in newly diagnosed prostate cancer patients. Thus, routinely measuring its values have the potential to be used while assessing disease severity and patient risk stratification, in addition to traditional clinical parameters such as the Gleason score, etc. As such, it may impact further treatment decisions and patient management. A future prospective study with a larger cohort and a longer follow-up period is required to validate the observed results.

## Figures and Tables

**Figure 1 cancers-15-01020-f001:**
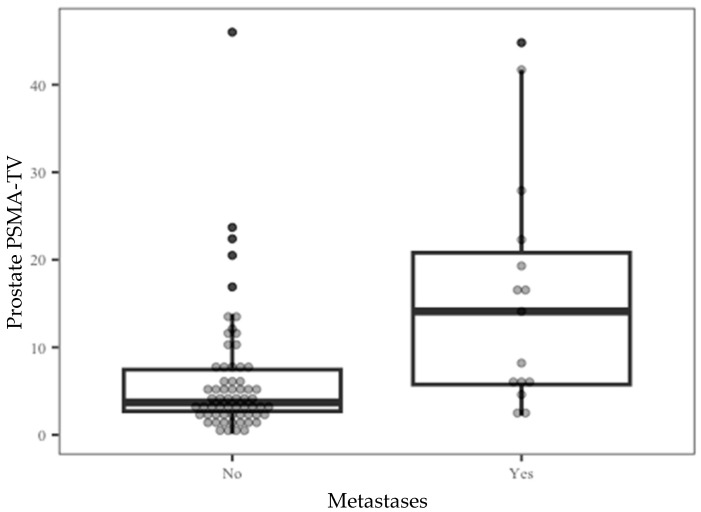
Prostate PSMA-TV distribution by the metastatic outcome.

**Figure 2 cancers-15-01020-f002:**
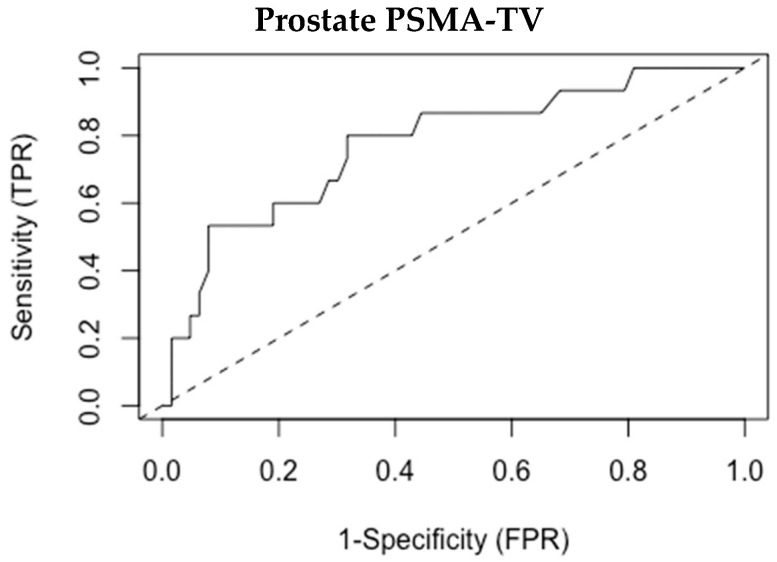
ROC analysis for prostate PSMA-TV.

**Table 1 cancers-15-01020-t001:** Descriptive statistics of the cohort (*n* = 85).

Parameter	Mean, [Standard Deviation]
Routinely used parameters	
Age (years)	72.07, [7.92]
Initial PSA	22.47, [6.79]
Gleason score	7.39, [0.88]
Follow-up length (months)	14.46, [5.5]
SUVmax in the prostate gland	12.19, [7.89]
SUVmean in the prostate gland	6.21, [4.31]
Advanced imaging parameters	
PSMA-TV	7.97, [9.57]
Prostate PSMA-TV	7.49, [7.54]
TL-PSMA	43.53, [74.12]
PSMA-TLQ	1.33, [1.48]

**Table 2 cancers-15-01020-t002:** Metastatic vs. non-metastatic cohort comparison.

Statistic	Metastatic Cohort		Non-Metastatic Cohort		*p*-Value
	Mean Results	Number of Patients	Mean Results	Number of Patients	
Routinely used parameters					
Age (years)	74.6	19	71.5	66	0.25
Initial PSA	18.5	9	9.3	57	0.06
Gleason score	8.2	15	7.2	61	<0.05
SUVmax	13.2	19	11.9	66	0.47
SUVmean	5.7	19	6.3	66	0.59
New imaging parameters					
PSMA-TV	21.3	12	5.5	66	<0.05
Prostate PSMA-TV	13.7	19	5.5	66	<0.05
TL-PSMA	87.7	19	34.7	66	0.1
PSMA-TLQ	2.2	19	1.1	66	0.05

**Table 3 cancers-15-01020-t003:** Prostate PSMA-TV values and presence of metastases.

Prostate PSMA-TV Value	Metastases + (*n* Patients)	Metastases − (*n* Patients)
≤8.1	6	55
>8.1	13	11

**Table 4 cancers-15-01020-t004:** Receiver Operating Characteristic (ROC) curve analysis.

Statistic	Value	95% CI
Sensitivity	80%	56% to 94.6%
Specificity	68.3%	56.2% to 78.9%
Positive Predictive Value (PPV)	37.5%	22.2% to 54.8%
Negative Predictive Value (NPV)	93.5%	84% to 98.3%

**Table 5 cancers-15-01020-t005:** Relationship between prostate PSMA-TV levels and metastases likelihood.

Prostate PSMA-TV Value	Metastasis + (*n* Patients)	% of Patients in Each Level	% of All Metastatic Patients	Metastasis − (*n* Patients)	% of Patients in Each Level	% of All Non-Metastatic Patients
Low (≤3.1)	2	7.1	10.5	26	92.9	39.4
Medium (3.2–6.4)	3	11.5	15.8	23	88.5	34.8
High (≥6.5)	14	45.2	73.7	17	54.8	25.8

## Data Availability

The data presented in this study are available in this article.

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
