# Peer review of "Advanced PSMA-PET/CT Imaging Parameters in Newly Diagnosed Prostate Cancer Patients for Predicting Metastatic Disease"

_cancers, 2023, doi:10.3390/cancers15041020_

Round 1
Reviewer 1 Report
Yechiel et al. evaluated the potential correlation between advanced PSMA PET parameters and metastatic
disease in patients undergoing PET for initial staging.
The authors found that higher prostate PSMA-TV predicts metastatic disease.
The study is well structured and well written. The above conclusion is novel as well as has the potential to impact clinical decision making.
Some minor adjustments are suggested:
1. The introduction is a bit too long, some of the background about the use of bone scintigraphy could be omitted.
2. Can the authors elaborate on the scan protocol? dosing, scan parameters, type of scanner etc.
3. The authors state that the PSMA-TV parameter is basically useless, so why even mention it?
4. Can the authors add the ROC curve? It would help the readers to better appreciate the utility of the 8.1 cutoff suggested. Overall it appears the prostate PSMA-TV is not such a great tool, as its best value is the NPV which is also not outstanding, a patient with prostate PSMA-TV of <8.1 still has 6.5% chance of having metastatic disease. However, it does show promise, considering that this is only a pilot study and I think that if the authors put this measure in the right prospective, it is a promising tool for the clinicians to use.
Author Response
- Comment: The introduction is a bit too long, some of the background about the use of bone scintigraphy could be omitted.
Answer: Thank you for your comments. We mentioned bone scintigraphy only in one paragraph, as this exam has been traditionally used, and still in use in many places for the assessment of skeletal involvement (metastasis presence, assessment after treatment etc.)
- Comment: Can the authors elaborate on the scan protocol? dosing, scan parameters, type of scanner etc.
Answer: In accordance with your suggestion we have elaborate on the scan protocol. The following paragraph was added to Methods:
18F-PSMA PET/CT acquisition and analysis: PET and contrast enhanced CT (when not contraindicated) were acquired consecutively from head to the mid-thigh using a PET/CT system (Discovery 690, GE Healthcare, Milwaukee, US), approximately 60 minutes after the injection on average of 296 MBq (8 mCi) 18F-PSMA. The following parameters were used for CT imaging: pitch 1.375:1, gantry rotation time 0.7 s, 120 kVp, automatically adjusted current in the range 100-650 mA, and a 2.5 mm slice thickness. A contrast enhanced CT scan was obtained 60 s after injection of 2 mL/kg of non-ionic contrast (Omnipaque 300; GE Healthcare). A PET scan followed in 3D acquisition mode for the same axial coverage. CT images were used for fusion with the PET data. PET images were reconstructed with CT attenuation correction using a 3D ordered subset expectation maximization (3D-OSEM).
All PET/CT studies were analyzed by a Nuclear Medicine specialist. All studies were reviewed retrospectively with knowledge of the patient’s clinical history and results of previous imaging studies.
- Comment: The authors state that the PSMA-TV parameter is basically useless, so why even mention it?
Answer: Indeed, PSMA-TV parameter did not contribute directly to our results, specifically regarding the ability of it to predict metastatic disease. However, we mentioned it in this article, as a parameter that was found to be significantly different between the two patient's groups and might emphasizes the differences between them. Although beyond the scope of this paper, future analysis may reveal some interesting findings regarding this basic advanced parameter, such as direct correlation with prostate TV, the differences between these two advances parameters, correlation with Gleason grade group etc.
- Comment: Can the authors add the ROC curve? It would help the readers to better appreciate the utility of the 8.1 cutoff suggested. Overall it appears the prostate PSMA-TV is not such a great tool, as its best value is the NPV which is also not outstanding, a patient with prostate PSMA-TV of <8.1 still has 6.5% chance of having metastatic disease. However, it does show promise, considering that this is only a pilot study and I think that if the authors put this measure in the right prospective, it is a promising tool for the clinicians to use.
Answer: In accordance with your suggestion, we added the ROC curve. Indeed, a patient with prostate PSMA-TV of <8.1 still has 6.5% chance of having metastatic disease, however, as you stated this is only a pilot study, and it may be still promising accompanied tool for the clinicians to use under the right prospective. We have added the following paragraph to Discussion:
In contrast, those with prostate PSMA-TV levels lower than 8.1 can be considered as having a smaller chance of developing metastatic lesions. However, since 10% of patients in this group still had a metastatic disease found in their initial imaging, it is by no means a 'magic bullet'. In a larger cohort, this model might miss a considerable number of patients with metastases, and therefore we believe that levels under 8.1 should not be used to rule-out the likelihood of a metastatic disease.
Reviewer 2 Report
Line 52-56: There are limitations when describing PSMA-PET/CT scans as the preferred staging modality, especially for primary stagings. High costs and the availability of the method are the main clinical limitations. However, the method showed the highest sensitivity and specificity in nodal metastases. Thus, in absence of prospective studies demonstrating survival benefit, caution should be taken, when assuming a preferrence.
Lines 115-123: Of the 85 described patients, there are only 74 attributed to a subgroup. 9 are missing.
Line 162: Why calculate a threshold for metastases in imagery diagnostic? A negative predictive value has always a lower clinical impact than a positive predictive value. However, the predictive value of these measurements would only be of clinical significance if they had a clinical prognostic value and not one for the imminent diagnostic result (findings in the imagery) of the diagnostic tool with the highest sensitivity and specifity available to date.
Lines 235-285: This is speculative. Sufficient follow-up data is missing. A follow-up of up to 20 months is a very short term for prostate carcinoma.
A comparison with more common staging methods for primary staging (CT-scan and szintigraphy) would add to the significance of the content of the paper.
Author Response
- Comment: Line 52-56: There are limitations when describing PSMA-PET/CT scans as the preferred staging modality, especially for primary stagings. High costs and the availability of the method are the main clinical limitations. However, the method showed the highest sensitivity and specificity in nodal metastases. Thus, in absence of prospective studies demonstrating survival benefit, caution should be taken, when assuming a preference.
Answer: Thank you for your comments. As you mentioned there are some limitations using PSMA-PET/CT scan. We changed the paragraph and added more details regarding these limitations. The following paragraph in now written in Discussion:
Hence, those patients with high-risk localized disease need a better diagnostic method for identifying distant metastasis. In recent years, PSMA-PET/CT is the preferred staging modality for both primary and recurrent prostate cancer due to its superior sensitivity and specificity [4-15], especially assessing extra-skeletal involvement such as nodal metastasis. The superior accuracy of PSMA-PET/CT over conventional imaging for staging in high-risk patients may allow identification of patients with otherwise occult distant metastatic disease and could facilitate individualized multimodal treatment concepts, especially in the setting of oligometastatic disease [4]….. However, it is important to mentioned some limitations regarding using this technology such as high costs and availability (geographical and temporal meaning), as well as this radiotracer pitfalls such as physiologic and other pathologic processes that can express PSMA, that imaging specialists should be familiar with.
- Comment: Lines 115-123: Of the 85 described patients, there are only 74 attributed to a subgroup. 9 are missing.
Answer: Thank you for drawing our attention to this mistake. After reexamination we found that the correct number of patients underwent active surveillance was 17, and not 8 as mistakenly was written. The corrected number was incorporated to the text.
- Comment: Line 162: Why calculate a threshold for metastases in imagery diagnostic? A negative predictive value has always a lower clinical impact than a positive predictive value. However, the predictive value of these measurements would only be of clinical significance if they had a clinical prognostic value and not one for the imminent diagnostic result (findings in the imagery) of the diagnostic tool with the highest sensitivity and specifity available to date.
Answer: Thank you for this comment. We mentioned the fact that patients with PSMA-TV of <8.1 still has the chance of having metastatic disease. However, we believe that this preliminary results of this relatively small pilot study, may serve as complimentary tool for the clinicians to use under the right prospective:
In contrast, those with prostate PSMA-TV levels lower than 8.1 can be considered as having a smaller chance of developing metastatic lesions. However, since 10% of patients in this group still had a metastatic disease found in their initial imaging, it is by no means a 'magic bullet'. In a larger cohort, this model might miss a considerable number of patients with metastases, and therefore we believe that levels under 8.1 should not be used to rule-out the likelihood of a metastatic disease.
- Comment: Lines 235-285: This is speculative. Sufficient follow-up data is missing. A follow-up of up to 20 months is a very short term for prostate carcinoma. A comparison with more common staging methods for primary staging (CT-scan and szintigraphy) would add to the significance of the content of the paper.
Answer: As you mentioned, a longer follow-up is needed as well as comparison to other staging methods. We mentioned those as part our study limitations:
"Our study has few limitations…. To this purpose and due to the ongoing nature of this disease, a longer follow-up period of time is required…. comparison to other common staging methods such as bone scintigraphy and CT scans and MR imaging may have added value to the significance of this study"
Moreover, we also mentioned this important comment in our conclusion:
"A future prospective study with a larger cohort and longer follow-up period of time is required to validate the results observed."
Round 2
Reviewer 2 Report
Changes are accepted.